# Engineered antimicrobial-derived peptides to manipulate mixed microbial systems

Vikas D. Trivedi,[1] James A. Van Deventer,[1,2] Nikhil U. Nair[1]

**ABSTRACT** Due to the complexity of microbial communities and coarseness of currently available manipulation techniques (e.g., transplantation and antibiotic treatment), it is often difficult to fully elucidate the interactions between members that define community structure and function from the top down. Thus, it is imperative to be able to observe and manipulate subpopulations within microbial communities to enable a fine-detail understanding of the full spectrum and mechanism of community functions. However, there is a technological gap that prevents targeted manipulation of subpopulations within intact microbial mixtures and communities. In this work, we develop molecular probes to manipulate specific subpopulations within multispecies microbial populations and validate these methods using model synthetic populations *in vitro*. We leverage the narrow spectrum of class II peptide bacteriocin (a bacterially synthesized antimicrobial peptide), pediocin PA-1, as a model to develop molecular probes. We first demonstrate the narrow-spectrum activity of pediocin and quantify its potency against a panel of bacteria. Next, we conjugate chemical handles on the bacteriocin and show that the binding spectrum is largely unchanged. Then, using truncated variants, also conjugated to chemical handles, we show functional non-bactericidal binders that largely maintain their specificity. Finally, with the unmodified and modified bacteriocins, we show that specific bacteria can be depleted through killing or cell sorting within a mixture of highly similar bacteria. While we developed the system with a single bacteriocin, we expect that the elucidated design rules may be applicable to a variety of natural bacteriocins to develop a generalized approach for manipulating specific bacterial members within a community. Development of such molecular probes would be transformative to advancing the mechanistic underpinnings of microbial community and microbiota structure-function relationships.

**IMPORTANCE** We demonstrate a novel approach for top-down manipulation of microbial co-cultures by leveraging the narrow-spectrum activity of a class IIa bacteriocin (bacterial-derived antimicrobial peptide). We expect this approach to be applicable to other microbial communities/microbiota when expanded to other bacteriocins and may prove to be an invaluable tool in studying structure-function relationships in microbial communities and engineering them for a variety of applications.

**KEYWORDS** chemical biology, synthetic ecology, co-cultures, top-down, AMP, peptide

There is increasing interest in understanding and leveraging bacterial communities to advance human, animal, plant, and environmental health as well as for biomanufacturing applications (1, 2). For example, gut microbes play a major role in regulating and modulating human metabolism (3–5), immune response (6–9), neural activity (10–13), etc., and directed alterations in microbiota function offer a powerful method to diagnose and combat physical and mental disorders and promote overall health (14, 15). However, the rules that underpin the functional relationships between members

**Peer Reviewer** Steph Smith, The University of North Carolina at Chapel Hill, Morehead City, North Carolina, USA

Address correspondence to Nikhil U. Nair, nikhil.nair@tufts.edu.

The authors declare no conflict of interest.

See the funding table on p. 12.

of microbial communities are poorly understood, which hinders our ability to rationally engineer them for beneficial outcomes. In general, we do not fully understand how mixtures of microbes form functional communities and what role each member plays over time, space, and in response to exogenous perturbations, etc. Knowing how microbial communities assemble and function is key to leveraging them for beneficial applications and will be transformational to basic, applied, and translational science and engineering (16).

Widely used sequencing-based approaches can correlate the presence of species with community phenotypes, but they are largely hypothesis generating. To test these hypotheses and to glean a deeper understanding of microbial communities, engineering tools and approaches are needed to non-destructively alter community composition. Although some insights can be gained by analyzing the behavior of synthetic few-member (often comprising <5 members) communities that are built from the bottom up, these approaches often do not scale to larger communities, nor can they be readily applied to natural ecosystems that may be spatiotemporally heterogeneous. Therefore, top-down approaches are needed (17). Antibiotics provide an obvious means to alter community composition with temporal control, but they are generally broad spectrum and are unable to target small subsets of bacteria. Alternatively, chemical fluorophores or fluorescent proteins can be used to tag species for real-time tracking studies (18). However, in these methods, cells must be modified *ex vivo* prior to introducing them to a community setting. This limits their utility to species that either have well-defined genetic tools (to express fluorescent proteins) and/or can be isolated from the remaining community members. There has been some very recent work to monitor and manipulate members within intact communities. For example, enzymatic activity present in subpopulations can be used to covalently tag species but cannot distinguish between similar species (19). Phages or their lysins can be used to eliminate specific members (20, 21), but lytic phages are not readily available for most bacteria or are hard to engineer for altered specificity, and lysins are labile and highly susceptible to rapid inactivation. Genetic alteration by *in situ* conjugation (22, 23) provides elegant methods to manipulate member functions within intact microbial communities, but it is difficult to limit them to a desired subset of species. To identify the functional role of individual species, their interactions with other members, and to follow how their functions may evolve over space and time, we need new reagents and methods to precisely manipulate select members. As gene editing tools (like CRISPR/Cas) have been invaluable in understanding how genetic composition controls cellular functions, analogous tools to study how community composition controls microbiota function could be transformative in advancing basic and translational microbiota science and engineering. However, there is a technological gap that limits targeted manipulation of subpopulations within communities.

In this work, we leverage natural and engineered bacterially derived antimicrobial peptides (AMPs) called bacteriocins as means to manipulate specific members within a mixed microbial system. Bacteriocins have been widely used as means to control contamination in food (24–28), as antimicrobials for human and animal health applications (29–34), and in understanding their role in interspecies competition (29, 35–41).

The prototypical class IIa bacteriocin pediocin PA-1 from *Pediococcus acidilactici* has been extensively studied for its anti-listerial activity and has been used to alter the gut environment to resist pathogen colonization and has been shown to be highly thermostable, potent, and narrow spectrum (29). We show that pediocin PA-1 can be used to specifically deplete a few members within a mixed community of highly similar bacterial species. We also demonstrate that pediocin PA-1 can be engineered to serve as a non-bactericidal binding reagent that specifically tags and manipulates a subset of species in a culture of phylogenetically similar species. Our research demonstrates the novel utility and promise of bacteriocins as a tool to modulate synthetic microbial communities, a concept that has heretofore been unexplored.

## RESULTS

### Purification of pediocin PA-1 yields a partially purified peptide with significant impurities

Pediocin PA-1 is known to display narrow-spectrum activity against only a few species that include *Lactobacillus coryniformis* B-4390 and various *Listeria* species (42). We re-tested the antimicrobial spectrum of the pediocin PA-1 producer strain, *Pediococcus acidilactici* UL5, using agar-well diffusion assays against a few representative members of lactic acid bacteria (LAB) and gram-negative, *Escherichia coli* DH5α as a control (Fig. S1). The effectiveness of the spent cell-free culture medium was evaluated by its ability to inhibit the growth of the indicator strains as measured by the zone of inhibition (ZOI). We found that the bioactivity of spent *P. acidilactici* UL5 supernatant was consistent with that of pediocin PA-1, as it only inhibited *L. coryniformis* (ZOI ~0.32 in.) and no other LAB (Fig. S1). Therefore, we conclude that the primary antimicrobial secreted by *P. acidilactici* UL5 was pediocin PA-1.

We then proceeded to clone the pediocin operon (*pedABCD* [43]) from *P. acidilactici* UL5 into the pSIP-P27 vector for heterologous expression in other lactic acid bacteria, viz., *Lactiplantibacillus plantarum* WCFS1, *Lactococcus cremoris* MG1363, and the well-established expression host, *E. coli* MG1655. The pediocin operon imparted anti-listerial activity to the *L. plantarum* culture supernatant but not to that of *L. cremoris* or *E. coli* (Fig. 1A). Based on the degree of bioactivity, we concluded that the level of recombinant biosynthetic pediocin PA-1 accumulated in *L. plantarum* culture was similar to that observed for *P. acidilactici* UL5 and was therefore not pursued further. Thus, for subsequent studies, we isolated pediocin PA-1 from *P. acidilactici* UL5 directly using the cell adsorption-desorption approach (Fig. S2). This method has been demonstrated previously as a viable technique to purify bacteriocins from spent media (44, 45). We found that using the producer strain *P. acidilactici* UL5 for adsorption-based purification instead of the target strain, *L. coryniformis*, was more convenient as it minimized the processing steps and hence reduced loss in final yield. Desorption attempted at pH 1.0 and 9.0 yielded similar peptide recovery, but the peptide recovered at the acidic pH was found to be more stable, which is consistent with prior data on pediocin PA-1 (44). We attempted to further purify the preparation using reverse-phase high performance liquid chromatography with minimal enhancement in purity but a significant loss in yield (Fig. S3). The peptide recovered after cell adsorption-desorption and dialysis was lyophilized for long-term storage to minimize any loss in activity. The peptide using this method was purified 189-fold with a specific activity of 9,434 $AU \cdot mg^{-1}$ as measured by agar-well diffusion method (Fig. 1B; Fig. S2A through C).

To determine the potency of the purified pediocin PA-1, we generated kill curves against a range of target bacteria (Fig. 1C). The concentration range tested was 3 pM to 3 µM following 10-fold serial dilutions. As expected, pediocin PA-1 displayed activity against *L. coryniformis* ($EC_{50}$ ~1.0 ± 0.5 nM) and *Listeria seeligeri* ($EC_{50}$ ~13.3 ± 3.5 nM). This is in approximate agreement with the published potency of pediocin PA-1 (46). Though the molecular mass of pediocin PA-1 is expected to be 4.5 kDa, we observed the partially purified peptide to migrate with an apparent mass of ~12.5 kDa (Fig. 1B). However, subsequent analysis by LC-MS yielded peaks of multiple m/z with the predominant peak at the expected molecular size of 4.6 kDa (Fig. S4). The purified pediocin PA-1, separated on denaturing but non-reducing SDS-PAGE (since this bacteriocin contains two crucial disulfide bonds), was subjected to agar-overlay assay using *L. seeligeri* as the indicator organism (Fig. S5). A single zone of clearing was observed, indicating that the purified peptide was indeed pediocin PA-1.

We conjugated the purified pediocin to Atto488 using 1-ethyl-3-(3-dimethylaminopropyl)carbodiimide−*N*-hydroxysuccinimide (EDC−NHS) chemistry (Fig. S6). We chose Atto488 due to its high fluorescence yield, high photostability, minimal aggregation, and because, with a net charge of −1, it does not significantly impact the overall cationic nature of pediocin. The Atto488-conjugated pediocin was dialyzed, and the excess dye

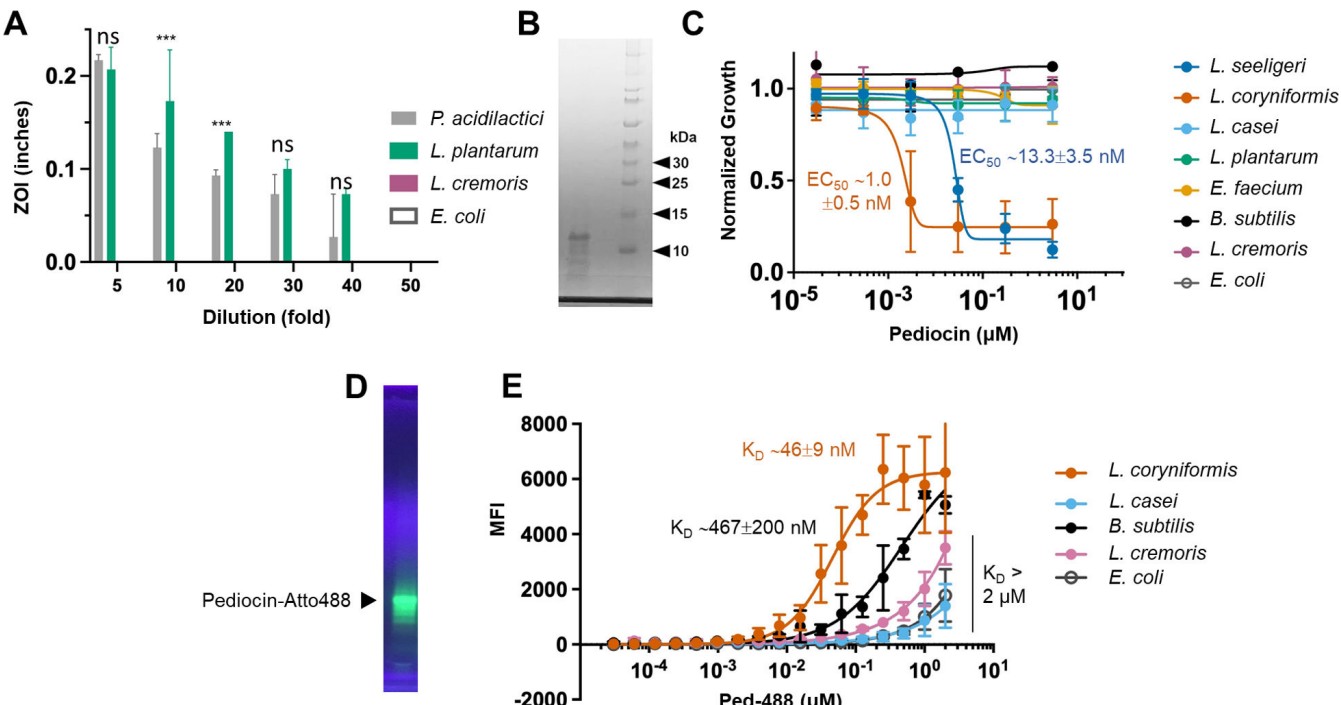

**FIG 1** Bioactivity spectrum of *P. acidilactici* UL5-derived biosynthetic pediocin PA-1. (A) Supernatant (10 µL) from *P. acidilactici* UL-5, or heterologously expressed in *L. plantarum*, *L. cremoris*, and *E. coli* was used as a source of pediocin PA-1 to assess bioactivity against *Listeria seeligeri*. Diameter of zone of inhibition (ZOI) measured as killing activity. *** is $P < 0.05$ for a two-tailed *t*-test; ns is not significant. (B) SDS-PAGE of partially purified pediocin PA-1 from *P. acidilactici*. Lane 1 is purified biosynthetic pediocin PA-1; lane 2 is empty; lane 3 is the ladder. (C) Kill-curve of pediocin PA-1 against target and non-target species. $EC_{50}$ for sensitive species, *L. coryniformis* and *L. seeligeri*, were $1.0 \pm 0.5$ nM and $13.3 \pm 3.5$ nM, respectively. (D) UV-fluorescence of pediocin-Atto488 (Ped-488) conjugate after separation on non-reducing SDS-PAGE. (E) Cell-binding curve of Ped-488 conjugate against the target and the non-target species. *E. coli* was used as a negative control. *L. coryniformis* demonstrated the highest affinity for pediocin with an apparent $K_D \sim 46 \pm 9$ nM. *Bacillus subtilis* showed $\sim 10\times$ higher apparent $K_D \sim 467 \pm 200$ nM. MFI, mean fluorescence intensity.

was removed using an acrylamide desalting column (1 kDa molecular weight cut-off [MWCO]; Fig. 1D). Since conjugation or labeling of AMPs to fluorescent probes has been reported to alter their antimicrobial activity in some cases (47), the pediocin-Atto488 conjugate (Ped-488) was also evaluated for its antimicrobial activity. The kill curve assay showed that pediocin maintained its antimicrobial activity after conjugation to Atto488. However, the $EC_{50}$ was increased by >100-fold relative to the unmodified peptide (Fig. S7). As for the binding affinity, among the strains tested, as expected, *L. coryniformis* showed the highest affinity and maximum median fluorescence for the conjugate (apparent $K_D \sim 46 \pm 9$ nM, 6,300 mean fluorescence intensity [MFI]; Fig. 1E; Table S1). The binding curve for the other tested microorganism did not reach saturation in the concentration range tested (up to 2 µM). *Bacillus subtilis* showed comparable binding for Ped-488 as indicated by the maximum MFI attained, albeit with lower affinity (apparent $K_D \sim 467 \pm 200$ nM). The other strains showed overall weak binding for Ped-488. Due to the broad binding seen by the conjugate preparation and the major non-specific band seen in the acrylamide gel, we decided not to move forward with the biosynthetic peptide and chemically synthesize it.

## Chemically synthesized pediocin PA-1 retains the bioactivity spectrum and potency

We ordered pediocin PA-1, which was chemically synthesized by Lifetein LLC (Somerset, NJ) and Biomatik Corporation (Kitchener, ON, Canada). Lifetein synthesized the peptide following a previously published method (48). This method relies on using solid-phase peptide synthesis via. Fmoc/tBu approach. Similar to the published method, the pediocin

synthesized for this work contained pseudoproline and C-terminal amide. We performed reducing SDS-PAGE analysis, and it displayed the expected molecular mass of ~4 kDa (Fig. 2A). We then confirmed its antimicrobial activity against using agar-well diffusion assay (Fig. 2B) and then quantified its $EC_{50}$ of synthetic pediocin PA-1 against *L. seeligeri* (11 ± 2.5 nM) and *L. coryniformis* (11 ± 5 nM; Fig. 2C). The difference in the $EC_{50}$ between the biosynthetic and synthetic Pediocin PA-1 could be due to the difference in the purity, active fraction, and other unknown interfering impurities. Nonetheless, the identical target spectrum and correct molecular mass gave us enough confidence to proceed with subsequent studies. We also compared the potency of pediocin PA-1 from Lifetein and Biomatik using *L. seeligeri* as an indicator organism (Fig. S7). The potency of the peptide from Lifetein ($EC_{50}$ ~40 ± 13 nM) was comparatively better than that from Biomatik ($EC_{50}$ ~193 ± 26 nM). This could be due to the difference in method adopted for synthesis and oxidation, stressing the need for characterization of synthetic antimicrobial peptides before proceeding with the intended study.

## Full-length pediocin PA-1 FITC-conjugate is more specific than the truncated peptides

Despite similar primary structures, the pediocin-like bacteriocins differ in their target cell specificity (i.e., they differ in their antimicrobial spectra) (49). By determining the target cell specificity of hybrid bacteriocins containing N- and C-terminal regions from different pediocin-like bacteriocins, it has been shown that the C-termini of these bacteriocins determine target cell specificity (49, 50). Mutational studies have been performed on pediocin PA-1 to elucidate the residue(s) responsible for its antimicrobial property, revealing residues Y2, G6, C9, C14, C24, W33, G37, and C44 to be critical for antimicrobial

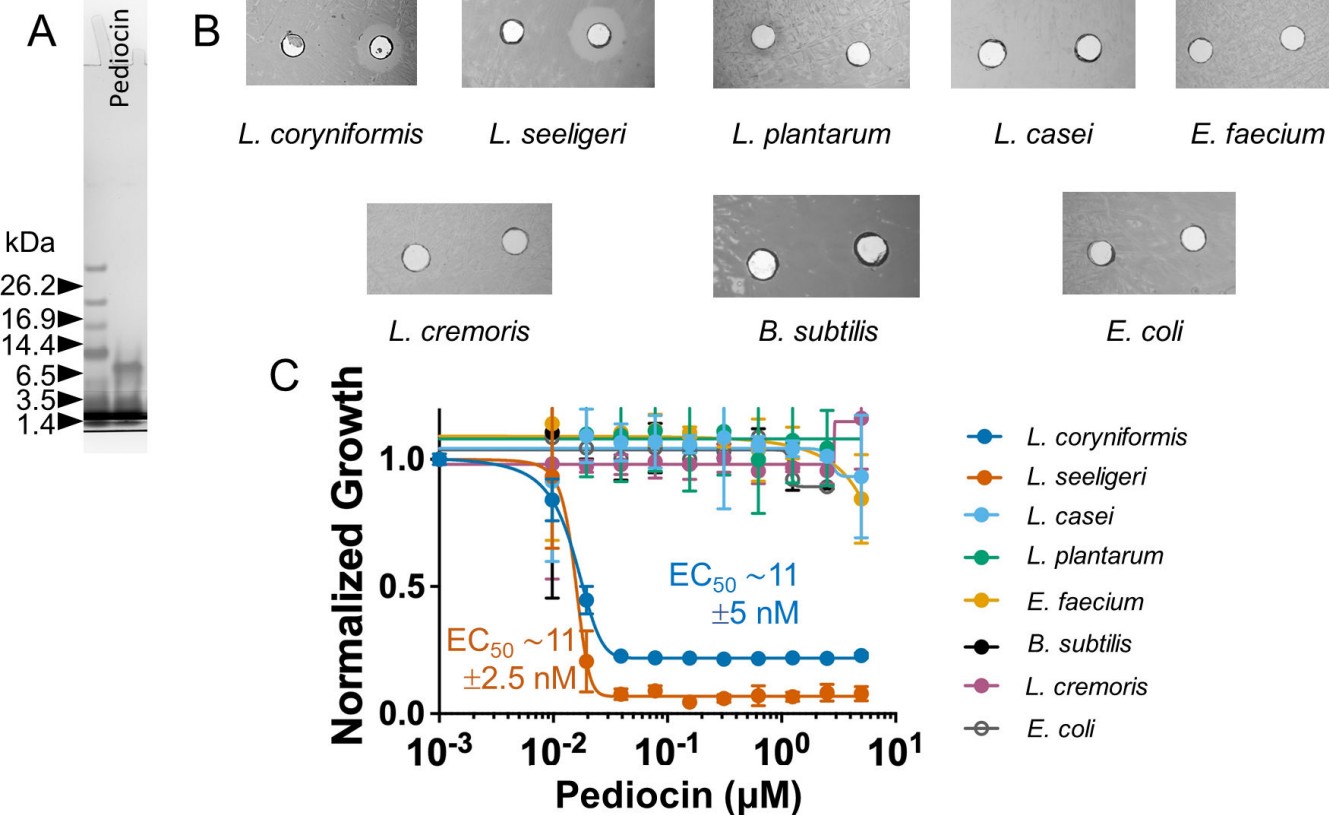

**FIG 2** Characterization of chemically synthesized pediocin PA-1. (A) SDS-PAGE analysis of pediocin PA-1 that was chemically synthesized by Lifetein, LLC shows a size as expected ~4 kDa. (B) Agar well diffusion assay of the synthetic pediocin PA-1 against a panel of bacteria indicated antimicrobial activity only against *L. coryniformis* and *L. seeligeri*. (C) Kill-curve of synthetic pediocin PA-1 against target and non-target species confirms high specificity and potency toward *L. coryniformis* and *L. seeligeri*, with similar $EC_{50}$ of ~11 ± 5 nM and 11 ± 2.5 nM, respectively.

activity (51). In addition, a 15-mer peptide fragment derived from the C-terminal half of pediocin PA-1 was found to specifically inhibit the bactericidal activity of pediocin PA-1 (52). Taking a cue from these studies, we proceeded to investigate whether fragments of pediocin PA-1 could serve as tags to specifically label pediocin-sensitive species without killing them. These could serve as tools to isolate target microorganisms from mixed samples. To this end, we synthesized fluorescein isothiocyanate (FITC)-conjugated versions of pediocin PA-1, a 15-mer (K20–A34) and an extended 30-mer fragment (S15–K43; Fig. S8). We first tested the bioactivity profile of FITC-conjugated pediocin and its fragments. We found that the antimicrobial activity of these peptides was completely abolished. Though the inactivity of 15-mer and 30-mer is expected, the loss of activity of pediocin-FITC conjugate could indicate the steric hindrance due to the fluorophore at the N-terminal, which is known to be critical for cell attachment (53). To test if this is due to the synthesis of the FITC-conjugate, we conjugated biotin, 7-nitrobenz-2-oxa-1,3-diazole (NBD), and Atto488 to the synthetic pediocin using EDC–NHS coupling (Fig. S6). These conjugates were all found to be inactive under the conditions tested (Fig. S7). We next tested the ability of 15-mer-FITC and 30-mer-FITC to inhibit the activity of the full-length unmodified synthetic pediocin but found them unable to do so, which could also be due to steric hindrance by the conjugated fluorophore (Fig. S9). These observations were concerning as they raised the possibility that any chemical modification of pediocin and its fragments would render them completely non-functional.

We then explored whether changing the pH altered the peptide binding properties. We observed that pediocin-FITC showed strong binding to *L. seeligeri* at all the tested pHs (4.0–7.4) but found surprisingly poor binding to *L. coryniformis* under acidic conditions (Fig. S10). In contrast, 30-mer-FITC and 15-mer-FITC demonstrated pH-dependent binding to *L. seeligeri*, with higher MFI at higher pHs. Based on the specificity and signal trade-off, pH 6.0 was chosen as the optimal condition to determine the apparent $K_D$ (Fig. 3). Pediocin-FITC showed specific binding to *L. seeligeri* (apparent $K_D$ = 82 ± 11 nM) with only *B. subtilis* showing modest binding at the highest concentration (30 µM) tested (Fig. 3A). The 30-mer-FITC conjugate did not show appreciable binding to *L. seeligeri* or the other species tested (Fig. 3B). Comparatively, the 15-mer-FITC conjugate showed significant binding to *L. seeligeri* (apparent $K_D$ = 2.8 ± 0.3 µM) and *L. cremoris* (apparent $K_D$ = 38 ± 4.7 µM; Fig. 3C). In addition, other species, *L. plantarum* and *Enterococcus faecium*, showed modest binding. This indicated that the specificity and affinity of binding drop significantly upon truncation and that the N-terminal residues may also play a role in determining the specificity and binding potency. Furthermore, stronger binding under acidic conditions suggests that protonated basic residues (e.g., lysine) in the N-terminus play an important role in binding interactions.

## Pediocin and its FITC conjugate can modulate a synthetic community in a targeted manner

We posited that the pediocin PA-1 characterized here could be a potential tool to modulate the abundance of specific species in an intact microbial community. To this end, we pursued two approaches: (i) using synthetic pediocin PA-1 to deplete specific species by killing and (ii) using FITC-conjugated pediocin PA-1 to deplete specific species by cell sorting. To deplete *L. seeligeri* by killing, we created a synthetic community of *Lacticaseibacillus casei*, *L. plantarum*, *L. cremoris*, *E. faecium*, and *L. seeligeri* and grew it for 2 h. We then split the culture into three parts to which we added 1 µM, 10 µM, or 0 (control) unmodified pediocin PA-1 (Lifetein), respectively. We observed that after 4 h (i.e., 6 h from inoculum), the proportion of *L. seeligeri* in the control (untreated) increased from 18 % to 60 %. Comparatively, in the culture where we added 1 µM or 10 µM pediocin PA-1, we observed that the proportion of *L. seeligeri* stayed unchanged at ~18% (Fig. 4A). This highlights the ability of pediocin PA-1 to modulate the synthetic community by inhibiting the growth of its target species. Next, we assessed the ability of FITC-conjugated pediocin (Ped-FITC) and its 15-mer version to deplete the target species through cell sorting. Again, we created a synthetic community containing *L. seeligeri* in

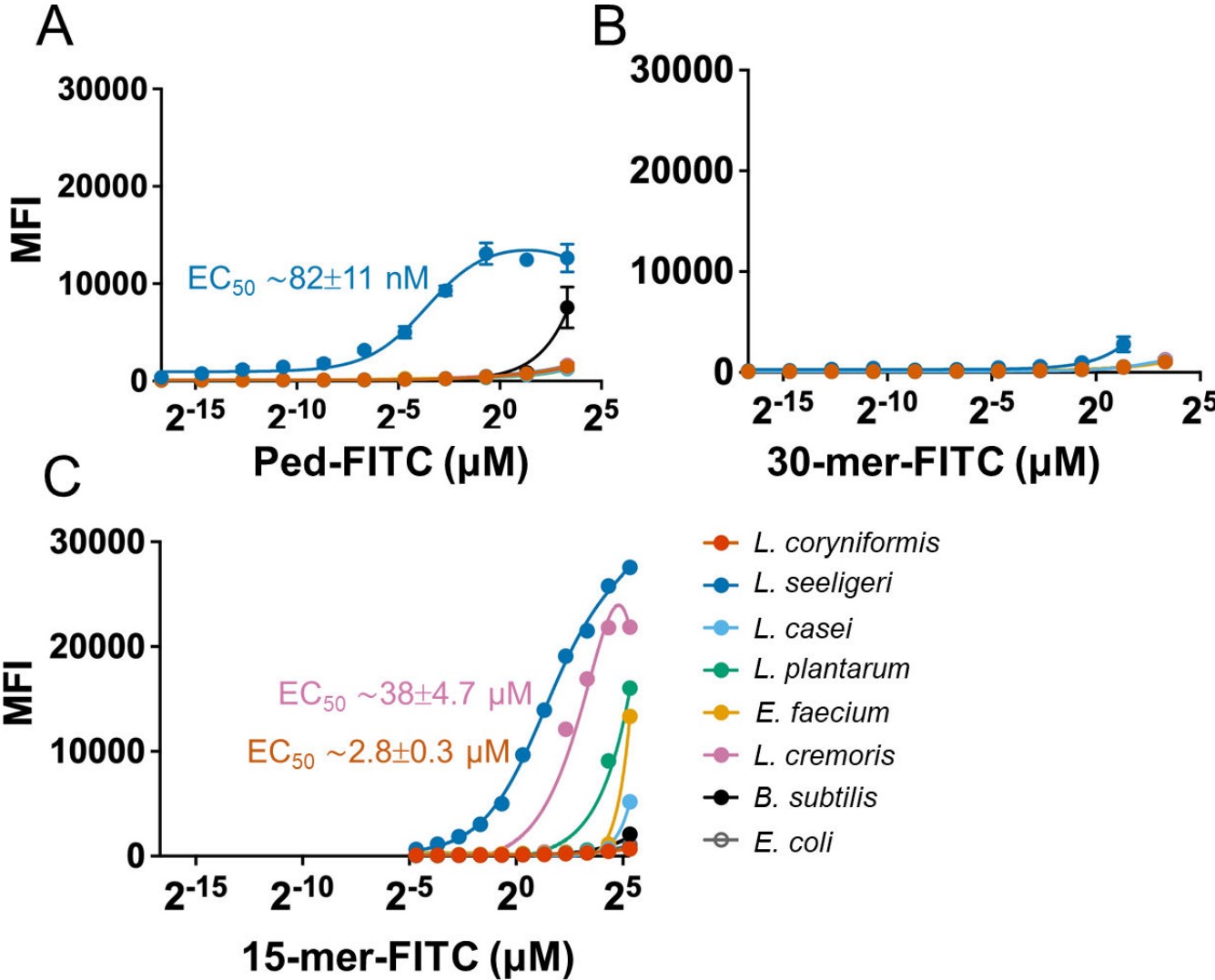

**FIG 3** Determining the binding constants of full-length and truncated pediocin PA-1 FITC conjugates. (A) The full-length pediocin is highly specific to *L. seeligeri* (apparent $K_D$ = 82 nM). (B) The 30-mer-FITC is unable to bind to any species tested. (C) The 15-mer-FITC binds most strongly to *L. corniformis* and modestly or weakly to all other species tested. The binding of the FITC peptide conjugate was determined by flow cytometry. All experiments were conducted at pH 6.0.

abundance and assessed the proportion before and after fluorescence-activated cell sorting (FACS) of FITC-labeled cells. We attempted to sequence the fluorescence-positive population but were unsuccessful due to poor DNA yield. However, the genomic DNA isolated from the fluorescence-negative population indicated depletion of more than 50% in the proportion of *L. seeligeri* from the synthetic community (Fig. 4B).

## DISCUSSION

In this study, we evaluated the antimicrobial spectrum, potency, and potential applications of pediocin PA-1 for manipulating microbial communities. Consistent with prior reports, the bioactivity of the crude supernatant of *P. acidilactici* UL5 was limited to *Listeria* species and *Lactobacillus corniformis*, with no detectable inhibition of other lactic acid bacteria or gram-negative *E. coli* (42). While we had some success using the purified, biosynthetic pediocin PA-1, chemical synthesis provided an alternate route to obtaining a product with expected properties—the correct molecular weight, narrow-spectrum activity, and $EC_{50}$ values in the low nanomolar range against *L. seeligeri* and *L. corniformis*.

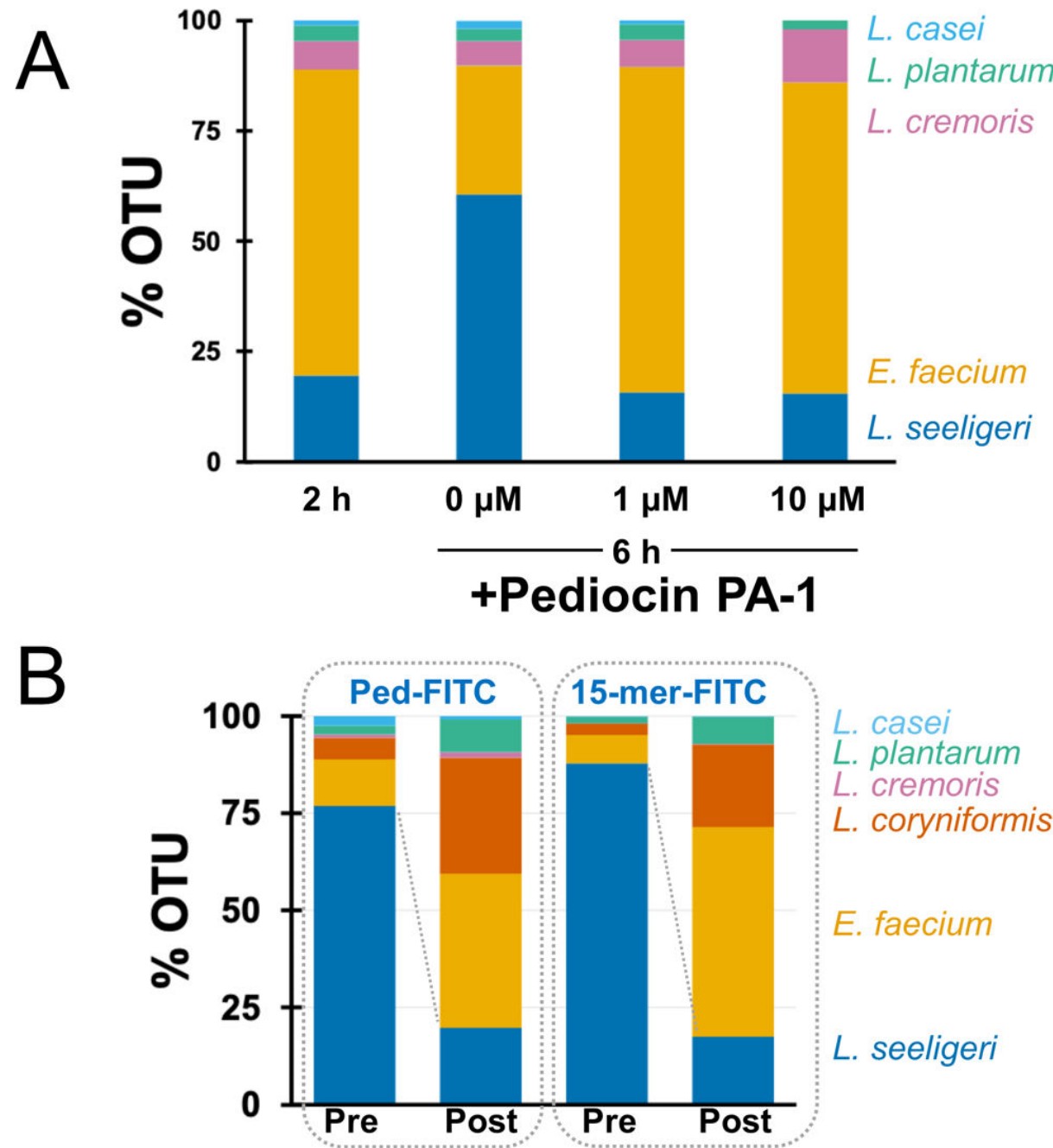

**FIG 4** Modulation of the synthetic community using pediocin PA-1 and its derivatives. (A) Treatment of the five-species synthetic community with unmodified full-length pediocin PA-1 results in targeted depletion of *L. seeligeri* relative to control. (B) Isolation of fluorescent-negative population by FACS of a five-species synthetic community after treatment with ped-FITC or 15-mer-FITC conjugates. In both cases, the V3–V4 region of 16S rDNA was amplified and sequenced using the Amplicon-EZ sequencing platform of Genewiz. OTU, operational taxonomic units.

Prior to this work, bactericidal activities of pediocin PA-1 and other narrow-spectrum bacteriocins have been well explored to control contamination in food (24–28), kill pathogens (29–34), and to study the role of probiotics and their derivatives on microbial communities (29, 35–41). However, they have not been well explored as tools to manipulate mixed microbial systems. Our results corroborate the established narrow specificity of pediocin PA-1 and demonstrate targeted killing in a mixed microbial system, highlighting their use for targeted depletion of a subset of species within communities. While other approaches have used phages (54–56), lytic enzymes (21, 57),

or intracellular toxins (including CRISPR/Cas) expression (22, 23, 58, 59) to deplete small numbers of species, they have limitations that our approach may overcome. For example, phages are not well known for many species, and identifying new phages is not trivial. Lytic enzymes are highly labile and readily inactivated or degraded. Intracellular delivery of toxic payloads poses significant hurdles in ensuring high-efficiency and targeted delivery and may require extensive engineering of conjugative, phage, or other gene-transfer systems. Furthermore, the addition of engineered species with payloads could drastically alter community dynamics, making them more suitable for applied and/or translational studies rather than those on the basics of microbial ecology. Comparatively, our peptide-based approach is simpler since bacteriocins against a wide variety of species have already been identified and can be mined from metagenomic sequences (60–64). Additionally, the ability to chemically synthesize peptide bacteriocins can make testing and validation quicker. Finally, peptide bacteriocins are generally less prone to denaturation, and since they kill cells extracellularly without transcription/translation steps, they can alter community composition more rapidly.

Beyond killing, we also demonstrated that modified peptides derived from pediocin PA-1 (whether conjugated to functional handles or truncated) maintain their target specificity while losing their bactericidal activity. Such reagents open the opportunity to manipulate mixed microbial systems from the top down in a non-lethal way to enable fundamental studies with extant microbial communities. This includes, but is not limited to, elucidating the roles of specific species in the establishment, evolution, and resilience of ecologies. Current approaches suffer from similar limitations as those described for killing—low specificity, slow response times, and complexity (19, 22, 65, 66).

Our approach is not without limitations; however, we found that the peptides show pH-dependent activity, which may affect their activity under uncontrolled conditions (e.g., *in situ*). Additionally, the peptides do not have absolute specificity, resulting in potential targeting of multiple species—such as various *Listeria* sp. and *L. coryniformis* by pediocin PA1. Our method development was supported by extensive published literature on the spectrum, potency, activity, and structure-function relationships of pediocin PA-1, which may not be readily available for other peptides. Thus, as with any method involving targeted manipulation of specific species, initial characterization will be needed prior to deployment.

Nonetheless, the potential utility of our approach is amplified by its overall simplicity. Since we used a naturally occurring bacteriocin and made small modifications, the approach could readily be adapted to other bacteriocins to manipulate other microbes and communities. With potential for multiplexing, this approach may provide a powerful path forward to advancing basic and applied science and engineering microbiota for biomedical, veterinary, agricultural, biomanufacturing, and environmental applications.

## MATERIALS AND METHODS

### Bacteria and plasmids used in this study

*Pediococcus acidilactici* UL-5, *Lactobacillus coryniformis* B-4390, *Weisella confusa*, *Lacticaseibacillus casei* B-1922, *Enterococcus faecium* NRRL B-2354, and *Listeria seeligeri* B33019 were obtained from USDA ARL-NRRL (Peoria, IL). *Lactiplantibacillus plantarum* WCSF1 was a kind gift from Prof. Michiel Kleerebezem (Wageningen University and Research, Wageningen, Netherlands), *Lactococcus cremoris* MG1363 was purchased from MoBiTec GmbH (Göttingen, Germany). The plasmid pSIP411 was a kind gift from Dr. Lars Axelsson (Nofima, Ås, Norway).

### Cloning of the pediocin PA-1 operon

Pediocin operon (*pedABCD*) was amplified from the gDNA of *P. acidilactici* UL5 and cloned into pSIP-P27 plasmid (67, 68) using the restriction-cloning method. The clone was verified by Sanger sequencing and used for heterologous expression in *E. coli* MG1655,

*L. plantarum* WCSF1, and *L. cremoris* MG1363. The plasmid was transformed into *E. coli* via the chemical heat shock method. *L. plantarum* and *L. cremoris* were transformed via electroporation.

## Bioactivity spectrum of pediocin PA-1

The antimicrobial spectrum of pediocin PA-1 was assayed using a kill curve. Cultures grown overnight were resuspended in fresh medium at an $OD_{600}$ of 0.05. $EC_{50}$ was determined by incubating cells with varying concentrations of pediocin in a 96-well plate, and growth was measured at 600 nm after 24 h.

## Purification of pediocin PA-1

*P. acidilactici* UL5 was grown on de Man, Rogosa, and Sharpe (MRS) broth for 20 h. The culture (2 L) was incubated for 30 min at 85°C with constant stirring, and the pH of the culture was adjusted to 6.0 using NaOH following the color on the pH strip. The pH-adjusted culture was incubated on ice for 4 h, following which cells were collected by centrifugation at $3,000 \times g$ at 4°C. Cells were washed thrice with ice-cold sodium-phosphate buffer (5 mM, pH 6.0) and resuspended in 100 mM NaCl (100 mL). After adjusting the pH of the cell suspension to 2.0, it was incubated at 4°C overnight to allow the desorption of pediocin. The supernatant was collected by centrifugation at $3,000 \times g$ for 20 min, dialyzed (Snakeskin dialysis tubing, 3 kDa MWCO, ThermoFisher Scientific, Carlsbad, CA) against water (1:10,000 dilution) at 4°C. Each dialysis step (1:10 dilution) was carried out for 6 h. Partially purified pediocin was lyophilized, resuspended in sodium-acetate buffer (pH 4.6, 100 mM), and stored at 4°C. Protein was estimated using Bradford reagent (VWR, Radnor, PA).

## Agar overlay assay

The antimicrobial activity of pediocin was assessed using an SDS-PAGE gel overlay assay (69). Protein samples were resolved on a 4%–20% acrylamide/bis-acrylamide SDS-PAGE gel, after which the gel was thoroughly washed with sterile water to remove residual SDS. The gel was then placed onto brain heart infusion (BHI) agar plates and over-laid with 0.8% BHI soft agar inoculated with *L. seeligeri*. Following incubation at 37°C overnight, plates were examined for zones of growth inhibition indicative of pediocin activity. The bactericidal activity of chimeric proteins was analyzed by polyacrylamide gel electrophoresis and gel overlay screening. Cell pellet and culture supernatant samples were first run on 16% acrylamide-bis-acrylamide–SDS gels. Subsequently, the gels were washed in sterile water for 3 h to remove SDS, placed on pre-poured tryptone glucose extract (TGE) agar plates, and covered with a 20 mL TGE soft-agar overlay containing *L. seeligeri* cells. The plates were incubated at 37°C overnight and examined for zones of growth inhibition associated with proteins in the samples.

## Conjugation of pediocin PA-1

Partially purified/synthetic pediocin PA-1 was conjugated to fluorophores (Atto488/NBD) using 1-ethyl-3-(3-dimethylaminopropyl)carbodiimide (EDC) chemistry. Briefly, 100 µL of fluorophore (2.5 mM, dissolved in phosphate-buffered saline [PBS]), 25 µL of EDC (20 mM in PBS), and 125 µL of dimethylformamide were incubated at room temperature (RT) in the dark for 10 min. Furthermore, 28 µL of *N*-hydroxysuccinimide (50 mM in PBS) was added and incubated at RT in the dark for another 10 min. At this point, 250 µL of pediocin (312 µM) was mixed with the activated Atto488 and incubated at RT in the dark for 2 h. The reaction was dialyzed (1:10,000) against PBS. The conjugated pediocin was then further separated from the unreacted dye using a polyacrylamide desalting column (1.8 kDa MWCO). The conjugated peptide was subjected to matrix-assisted laser desorption/ionization–time of flight to determine mass and SDS-PAGE followed by visualization under UV for fluorescence signal.

## Binding assay

The binding affinity and avidity of pediocin to various gram-positive and gram-negative bacteria were studied using flow cytometry. Late log-phase cells were washed in the sodium phosphate buffer (5 mM, pH 6.0) and resuspended at an $OD_{600}$ of $10^{-3}$ in sodium-acetate buffer (25 mM, pH 4.0), sodium phosphate buffer (5 mM, pH 6.0), or sodium phosphate buffer (5 mM, pH 7.4) containing 1% bovine serum albumin (BSA) and 100 mM NaCl. Cells were incubated with varying concentrations of the conjugated peptide at 4°C for 30 min. The unbound peptide was removed by centrifugation, and the pellet was resuspended in its respective binding buffer containing 1% BSA. Flow cytometry was performed on an Attune NxT flow cytometer (Life Technologies/Thermo-Fisher Scientific, Carlsbad, CA). Data ($\geq 10^4$ events were recorded) were recorded using a green laser (488 nm) at a voltage of 360 V, and gating was performed on an SSC-A versus SSC-H plot to reduce false events.

## Culturing of the mixed community

After overnight growth of each microbial member in their preferred media (MRS for *Lacticaseibacillus casei*, *L. plantarum*, and *Enterococcus faecium*; GM17 for *L. cremoris*; BHI for *Listeria seeligeri*), cells were washed thoroughly in sodium phosphate buffer (5 mM, pH 6.0). Equal colony-forming units for each species were mixed and used to inoculate BHI medium for co-culture at 30°C.

## Binding assay in the mixed community

Late log-phase cells were washed in the sodium phosphate buffer (5 mM, pH 6.0) and resuspended at an $OD_{600}$ of 0.1 in sodium phosphate buffer (5 mM, pH 6.0) in the presence of 1% BSA and 100 mM NaCl. Cells were mixed in equimolar ratios and incubated with 40 µM of Ped-FITC at 4°C for 30 min. The unbound peptide was removed by centrifugation, and the pellet was resuspended in sodium-phosphate buffer (5 mM, pH 6.0) containing 1% BSA. FACS was performed on BioRad S3e (BioRad, Hercules, VA). An attempt was made to sort at least $10^5$ cells; the FITC-positive and negative cells were sorted, and cells were collected by centrifugation to extract gDNA (genomic DNA) and amplify the 16S V3–V4 region of rDNA (ribosomal RNA encoding DNA).

## Killing assay in the mixed community

$OD_{600}$ of overnight cultures grown in their respective media was determined and mixed in equimolar ratios. The synthetic community was inoculated in BHI media (20 mL), and 5 mL of the sample was stored as a time-zero sample ($t = 0$). After 2 h, 5 mL of the sample was collected at $t = 2$ h, and an additional 5 mL sample was allowed to grow separately for the duration of incubation as a control. Pediocin (100 µM) was added to the rest of the culture, which was then allowed to grow for an additional 2 h. At this point, the control and pediocin-treated cultures were harvested. All the cell pellets were stored at −20°C until further processing. gDNA from the pellets was isolated, and the 16S V3–V4 rDNA region was amplified and barcoded via primers. The barcoded samples were pooled into six groups and submitted to Genscript for Amplicon-EZ sequencing.

## Data processing

16S rDNA of all cultures was independently sequenced and verified. The verified FASTA sequences were imported into Geneious Prime and consolidated into a single list to create a reference file. The FASTA sequences from the amplicon sequencing were imported into Genious Prime and demultiplexed using the pre-assigned barcodes. The demultiplexed samples were then mapped to the reference file using BowTie2 to determine the counts of each species (70). The counts were exported into an Excel sheet for further processing and plotting.

## ACKNOWLEDGMENTS

We thank all past and present members of the Nair lab for their helpful discussion, especially Dr. Todd C. Chappell, Dr. Zachary J. S. Mays, and Dr. Josef R. Bober.

This work was funded by NIH grant #DP2HD091798, NSF grant #1,935,354, and a Tufts Launchpad | Accelerator grant (to N.U.N.).

N.U.N. and V.D.T. conceived and designed the research project. V.D.T., J.A.V.D., and N.U.N. wrote and edited the manuscript. V.D.T. performed the experiments. V.D.T., J.A.V.,D. and N.U.N. analyzed the data. All the authors have reviewed the manuscript and approved it for submission.

## AUTHOR AFFILIATIONS

[1]Department of Chemical and Biological Engineering, Tufts University, Medford, Massachusetts, USA
[2]Department of Biomedical Engineering, Tufts University, Medford, Massachusetts, USA

## PRESENT ADDRESS

Vikas D. Trivedi, Center of Excellence for Data-Driven Discovery, Department of Structural Biology, St. Jude Children's Research Hospital, Memphis, Tennessee, USA

## AUTHOR ORCIDs

Nikhil U. Nair ⓘ http://orcid.org/0000-0001-7737-1385

## FUNDING

| Funder | Grant(s) | Author(s) |
|---|---|---|
| National Institutes of Health | DP2HD091798 | Nikhil U. Nair |
| National Science Foundation | 1935354 | Nikhil U. Nair |
| Office of the Vice Provost for Research, Tufts University | Tufts Launchpad Accelerator | Nikhil U. Nair |

## AUTHOR CONTRIBUTIONS

Vikas D. Trivedi, Conceptualization, Formal analysis, Investigation, Methodology, Writing – original draft, Writing – review and editing | James A. Van Deventer, Formal analysis, Supervision, Writing – review and editing | Nikhil U. Nair, Conceptualization, Funding acquisition, Methodology, Project administration, Resources, Supervision, Writing – original draft, Writing – review and editing

## DATA AVAILABILITY

The raw sequencing files can be accessed at NCBI SRA BioProject number PRJNA1380370. Additional details on specific sequences submitted are described in Data S1.

## ADDITIONAL FILES

The following material is available online.

### Supplemental Material

**Data S1 (Spectrum03215-25-s0001.xlsx).** Details on samples and 16S rDNA sequence information submitted to NCBI SRA BioProject number PRJNA1380370.
**Supplemental material (Spectrum03215-25-s0002.docx).** Fig. S1 to S10; Table S1.

Open Peer Review

**PEER REVIEW HISTORY (review-history.pdf).** An accounting of the reviewer comments and feedback.

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
