## [Reviewer comments · Microbiology Spectrum]

Microbiology Spectrum

Engineered antimicrobial-derived peptides to manipulate mixed microbial systems

Vikas Trivedi, James Van Deventer, and Nikhil Nair

Corresponding Author(s): Nikhil Nair, Tufts University

Review Timeline:

Submission Date:	October 6, 2025
Editorial Decision:	October 23, 2025
Revision Received:	January 6, 2026
Editorial Decision:	January 27, 2026
Revision Received:	January 28, 2026
Accepted:	February 10, 2026

Editor: Sergey Zotchev

Reviewer(s): Disclosure of reviewer identity is with reference to reviewer comments included in decision letter(s). The following individuals involved in review of your submission have agreed to reveal their identity: Steph Smith (Reviewer #2)

Transaction Report:

DOI: <https://doi.org/10.1128/spectrum.03215-25>

Re: Spectrum03215-25 (**Engineered antimicrobial-derived peptides to manipulate mixed microbial systems**)

Dear Prof. Nikhil U Nair:

Thank you for the privilege of reviewing your work. Below you will find my comments, instructions from the Spectrum editorial office, and the reviewer comments.

Revision Guidelines

Sincerely,
Sergey Zotchev
Editor
Microbiology Spectrum

Reviewer #1 (Public repository details (Required)):

Results from the 16S rDNA sequencing for killing experiments

Reviewer #1 (Comments for the Author):

The paper entitled « Engineered antimicrobial-derived peptides to manipulate mixed microbial systems » by Trivedi and colleagues presents a comprehensive investigation into the purification, fluorescent conjugation and use for killing or labeling specific species of the class II peptide bacteriocin, Pediocin PA-1. The idea beyond is the development of efficient methods to kill and label specific species in complex communities.

While the paper is well written and can be followed easily. The novelty of the proposed strategy can be challenged. The introduction of this manuscript is short in exposing the past studies that used bacteriocins in, for instance, the context of food chain/cheese contamination by *Listeria monocytogenes* (for example lacticin 3147 and nisin maybe?). The introduction will also be improved in citing the work on the role of bacteriocins in shaping community structure through competitive interactions (see the work of Kevin R. Foster).

The discussion section is not a discussion, but merely a summary of the results, as evidenced by the absence of citations from other works that would provide further context for the study presented. The discussion would be really enriched in explaining in which environment such methodologies could be used: in the food chain, in the soil to promote certain beneficial plant-growth-promoting bacteria, in clinics. One would thus be very interested to get information on what is known, for instance in clinics, of already identified bacteriocins specifically targeting important human pathogens. This discussion would also be welcoming to expose the limitations of the use of bacteriocins in complex communities (that can sometimes contain several hundreds of species) such as the difficulty to formally demonstrate their specificity in such complex communities, the questioning about the best methodologies to deliver the bacteriocins, the pro and cons of using purified bacteriocins as compare to the producing micro-organisms directly, etc etc etc.

In addition to these conceptual missing aspects there are other questions that should be answered or aspects that should be clarified:

- It is easier for the reviewers to highlight concerns that should be considered when providing page and line numbers.
- Why not using *L. seeligeri* in Figure S1 while using it in Figure 1A ?
- *L. coryniformis* being more sensitive to Pediocin than *L. seeligeri* it could be good to use it for the test of Figure 1A.
- in Figure 1A, please provide statistical analysis to be able to say that the activity of the supernatant of *L. plantarum* is similar to the one of *P. acidilactici*, because it looks like of bit higher instead.
- One would have appreciated in Figure 1A or elsewhere to see the activity of control strains such as *L. plantarum* without the plasmid containing the Pediocin synthetic genes or with the empty plasmids, as well as strains of *P. acidilactici* where this operon is inactivated (since genetic tools have been developed for this bacterium).
- In Figure 1C and 1E, please change the yellow color for *B. subtilis* since it is barely visible.
- For the conjugation of Pediocin, one would appreciate to have information on the efficacy and percentages of conjugation since the presence of non-conjugated pediocin in the used solution could strongly impact on the killing efficacy and binding measurements.
- For figure S7 it is said that « However, the EC50 was manyfold lower relative to the unmodified peptide ». Please provide real numbers characterizing the difference of activity.
- Could the authors clarified in the text that they called the purified version of Pediocin from *L. plantarum* a Biosynthetic version of it ?
- For figure 2C, it is said : « and then quantified its EC50 of synthetic pediocin PA-1 against *L. seeligeri* (2.5 nM) and *L. coryniformis* (4.7 nM) ». I think this is wrong. The EC50s are 11 nM +/- 2.5 and 5 nM.
- Can the authors comment on the fact that the EC50 of the synthetic Ped is 10 fold higher than the purified one for *L. coryniformis* but the same for *L. seeligeri*.
- In Figure S8 it could be helpful to indicate the localisation of the N-ter and C-ter part of the peptides, as well as, the parts described as responsible for the toxic vs binding activities.
- The Kd measurement of the Pediocin-FITC has been done in specific pH conditions. How, for example, the Kd of 82 nM to *L. seeligeri* compares to the one of the non conjugated Pediocin performed in the exact same conditions ?
- In relation with the experiments on synthetic communities, one can ask what would be happen if in the mix other *Listeria* species would be included. One of the main limitation of the use of bacteriocins is the fact that they can sometime target close species (which, by the way, question on their use in very complex communities where numerous species from the same genera are cohabiting).
- Also could the authors comment on their choice of using a 5 species community when testing killing activity while using a 6 species community (with the addition of *L. coryniformis*) when testing binding activity.
- Why *L. coryniformis* is not depleted (but instead is enhanced) is the fluorescent-negative population since it is a target of Pediocin (Figure 4B) ?
- Since there is an important debate on the use of probiotics versus their active compounds to control certain pathogenic bacteria, it would be very interesting to see how does *P. acidilactici* (and ideally its pediocin isogenic mutant) perform in killing experiments with complex communities compared to the purified or synthetic version of Pediocin.
- Could Table S1 provide the binding of pediocin to *L. seeligeri* ?
- For the killing assay in the mixed community, how accurate and quantitative enough is the use of 16S DNA PCR and

sequencing ? What about the presence of extracellular DNA coming from dead cells ? Did the authors considered ddPCR and the use of for exemple PMA technology to ensure the absence of amplification from dead cells ?

Reviewer #2 (Comments for the Author):

This is an exciting manuscript that presents a valuable tool for the microbiome research community. If anything, the work may be underselling the broader potential of this approach. I look forward to implementing this technique in my own research and expect others will be equally enthusiastic. The authors may wish to further emphasize the proof-of-concept nature of this work and its potential for diverse applications.

Overview

This manuscript presents a clever approach for targeted microbiome manipulation using narrow-spectrum class II peptide bacteriocins. The authors demonstrate this technique using pediocin PA-1 as a proof-of-concept, and the methodology could be readily adapted to other natural bacteriocins for various applications.

Major Comments

1. The Discussion section would benefit from addressing how this bacteriocin-based approach compared to other methods for specific microbiome labeling or manipulation. Consider discussing the relative advantages, limitations, and potential use cases of this technique compared to alternative approaches such as CRISPR-based methods.
2. The Discussion should more strongly emphasize the significance of this technique's dual capability: (1) specifically labeling pediocin-sensitive species without killing them, and (2) selectively eliminating specific community members. Both applications have compelling use cases!
3. Please address potential limitations when adapting this technique to other systems. Specifically, successful implementation in an undefined community requires: (1) identification of an appropriate bacteriocin that targets the organism of interest, and (2) comprehensive knowledge of community composition and bacteriocin-specificity to predict off-target effects. While this doesn't diminish the utility of this approach, it does mean that the technique would require quite a bit of consideration to transfer to new systems.

Minor Comments

Abstract

- Two consecutive sentences begin with "Finally". Please rephrase one to improve readability.

Introduction, Paragraph 2

- Is the paragraph intended to begin with "But"?
- Consider revising "sequencing-based studies" to "sequencing-based approaches."
- The rationale for needing additional microbiome manipulation tools is valid, but the connection could be clarified. Please explicitly explain how tools enabling targeted community alteration support the testing of hypotheses generated by traditional/sequence-based approaches.
- Final sentence: Consider changing "prevents targeted manipulation" to "limits targeted manipulation."

Results, Paragraph 2

- The authors state they cloned the "pediocin operon" into pSIP-P27 but do not define how this operon was delineated. Has this operon been functionally characterized previously? If so, please provide a reference. If not, please describe how the cloned region was determined.

Methods

- The description of agar overlay assays is missing from the Methods section. Please include this protocol.

Responses to reviewer are in blue. Updated manuscript text is in red.

Reviewer #1

1. **(Public repository details (Required)):**

Results from the 16S rDNA sequencing for killing experiments

We have deposited the sequences to NCBI SRA. The BioProject number is PRJNA1380370. We have added the following statement to the Method sections:

The raw sequencing files can be accessed at NCBI SRA BioProject number PRJNA1380370.

Additional details on specific sequences submitted are described in Supplemental File 1.

The paper entitled « Engineered antimicrobial-derived peptides to manipulate mixed microbial systems » by Trivedi and colleagues presents a comprehensive investigation into the purification, fluorescent conjugation and use for killing or labeling specific species of the class II peptide bacteriocin, Pediocin PA-1. The idea beyond is the development of efficient methods to kill and label specific species in complex communities.

While the paper is well written and can be followed easily. The novelty of the proposed strategy can be challenged.

2. The introduction of this manuscript is short in exposing the past studies that used bacteriocins in, for instance, the context of food chain/cheese contamination by *Listeria monocytogenes* (for example lacticin 3147 and nisin maybe?). The introduction will also be improved in citing the work on the role of bacteriocins in shaping community structure through competitive interactions (see the work of Kevin R. Foster).

We have edited the introduction to include additional references to the use of bacteriocins in combating food contamination, infections, as well as fundamental studies on studying how they shape mixed microbial systems. The last paragraph of the Introduction has been edited as follows:

In this work, we leverage natural and engineered bacterially-derived antimicrobial peptides (AMPs) called bacteriocins as means to manipulate specific members within a mixed microbial system. Bacteriocins have been widely used as means to control contamination in food [24–28], as antimicrobials for human and animal health applications [29–34], and in understanding their role in interspecies competition [29, 35–41].

The prototypical class IIa bacteriocin pediocin PA-1 from *Pediococcus acidilactici* has been extensively studied for its anti-Listerial activity and has been used to alter the gut environment to resist pathogen colonization and has been shown to be highly thermostable, potent, and narrow spectrum [29]. We show that pediocin PA-1 can be used to specifically deplete few members within a mixed community of highly similar bacterial species. We also demonstrate that pediocin PA-1 can be engineered to serve as a non-bactericidal binding reagent that specifically tags and manipulates a subset of species in a culture of phylogenetically similar species. Our research demonstrates the novel utility and the promise of bacteriocins as a tool to modulate synthetic microbial communities, a concept that has heretofore not been explored.

3. The discussion section is not a discussion, but merely a summary of the results, as evidenced by the absence of citations from other works that would provide further context for the study presented.

The discussion would be really enriched in explaining in which environment such methodologies could be used: in the food chain, in the soil to promote certain beneficial plant-growth-promoting bacteria, in clinics. One would thus be very interested to get information on what is known, for instance in clinics, of already identified bacteriocins specifically targeting important human pathogens. This discussion would also be welcoming to expose the limitations of the use of bacteriocins in complex communities (that can sometimes contain several hundreds of species) such as the difficulty to formally demonstrate their specificity in such complex communities, the questioning about the best methodologies to deliver the bacteriocins, the pro and cons of using purified bacteriocins as compare to the producing micro-organisms directly, etc etc etc.

Thank you for this feedback. We have now edited the Discussion to add additional details on the potential implications and applications of this work.

In **this** study we evaluated the antimicrobial spectrum, potency, and potential applications of pediocin PA-1 **for manipulating microbial communities**. Consistent with prior reports, the bioactivity of the crude supernatant of *P. acidilactici* UL5 was limited to *Listeria* species and *Lactobacillus coryniformis*, with no detectable inhibition of other lactic acid bacteria or Gram-negative *E. coli* [42]. **While we had some success with using the purified, biosynthetic pediocin**

PA-1, chemical synthesis provided an alternate route to obtaining a product with expected properties – correct molecular weight, narrow-spectrum activity, and EC₅₀ values in the low nanomolar range against *L. seeligeri* and *L. coryniformis*.

Prior to this work, bactericidal activities of pediocin PA-1 and other narrow-spectrum bacteriocins have been well-explored to control contamination in food [24–28], kill pathogens [29–34], and in studying the role of probiotics and their derivatives on microbial communities [29, 35–41]. But they have not been well-explored as tools to manipulate mixed microbial systems. Our results corroborate the established narrow-specificity of pediocin PA-1 and demonstrated targeted killing in a mixed microbial system, highlighting their use for targeted depletion of a subset of species within communities. While other approaches have used phages [53–55], lytic enzymes [21, 56], or intracellular toxins (including CRISPR/Cas) expression [22, 23, 57, 58] to deplete small numbers of species, they have limitations that our approach may overcome. For example, phages are not well-known for many species and identifying new phages is not trivial. Lytic enzymes are highly labile and readily inactivated or degraded. Intracellular delivery of toxic payloads poses significant hurdles in ensuring high-efficiency and targeted delivery and may require extensive engineering of conjugative, phage, or other gene-transfer systems. Further, addition of engineered species with payloads could drastically alter community dynamics, making them more suitable for applied and/or translational studies rather than those on basics of microbial ecology. Comparatively, our peptide-based approach is simpler since bacteriocins against a wide-variety of species have already been identified and can be mined from metagenomic sequences [59–63]. Additionally, the ability to chemically synthesize peptide bacteriocins can make testing and validation quicker. Finally, peptide bacteriocins are generally less prone to denaturation and since

they kill cells extracellularly without transcription/translation steps, they can alter community composition more rapidly.

Beyond killing, we also demonstrated that modified peptides derived from pediocin PA-1 (whether conjugated to functional handles or truncated) maintain their target specificity while losing their bactericidal activity. Such reagents open the opportunity to manipulate mixed microbial systems from the top-down in a non-lethal way to enable fundamental studies with extant microbial communities. This includes, but is not limited to, elucidating roles of specific species in establishment, evolution, and resilience of ecologies. Current approaches suffer from similar limitations as those described for killing – low specificity, slow response times, and complexity [19, 22, 64, 65].

Our approach is not without limitations, however. We found that the peptides show pH-dependent activity, which may affect their activity under uncontrolled conditions (e.g., in situ). Our method development was supported by extensive published literature on the spectrum, potency, activity, and structure-function relationships of pediocin PA-1, which may not be readily available for other peptides. Thus, as with any method involving targeted manipulation of specific species, initial characterization will be needed prior to deployment.

Nonetheless, the potential utility of our approach is amplified by its overall simplicity. Since we used a naturally-occurring bacteriocins and made small modifications, the approach could readily be adapted to other bacteriocins to manipulate other microbes and communities. With potential for multiplexing, this approach may provide a powerful path forward to advancing basic and applied

science and engineering microbiota for biomedical, veterinary, agricultural, biomanufacturing, and environmental applications.

4. In addition to these conceptual missing aspects there are other questions that should be answered or aspects that should be clarified:
- It is easier for the reviewers to highlight concerns that should be considered when providing page and line numbers.

We apologize – we in many other cases, we found journals automatically include page and line number into manuscripts. Since that did not happen here, we have added them.

- Why not using *L. seeligeri* in Figure S1 while using it in Figure 1A ?

Unfortunately, we did not conduct these experiments concurrently, which is why there is no direct correspondence between Figure 1A and S1. However, we believe our assertions and conclusions for this study remain unchanged, especially when combined with data on *L. seeligeri* in Figure S3C.

- *L. coryniformis* being more sensitive to Pediocin than *L. seeligeri* it could be good to use it for the test of Figure 1A.

Although you are technically correct, we chose *L. seeligeri* since pediocin PA-1 is primarily characterized by its anti-listerial activity. In the light of this well-accepted property, we think that *L. seeligeri* is likely the optimal choice for this assay.

- in Figure 1A, please provide statistical analysis to be able to say that the activity of the supernatant of *L. plantarum* is similar to the one of *P. acidilactici*, because it looks like of bit higher instead.

We have added a statistical test. The updated Figure 1A is below:

Figure 1. Bioactivity spectrum of *P. acidilactici* UL5-derived biosynthetic pediocin PA-1. (A) Supernatant (10 μ L) from *P. acidilactici* UL-5, or heterologously expressed in *L. plantarum*, *L. cremoris*, and *E. coli* was used as source of pediocin PA-1 to assess bioactivity against *L. seeligeri*. Diameter of

zone of inhibition (ZOI) measured as killing activity. *** is $p < 0.05$ for a two-tailed t-test; ns is not significant.

- One would have appreciated in Figure 1A or elsewhere to see the activity of control strains such as *L. plantarum* without the plasmid containing the Pediocin synthetic genes or with the empty plasmids, as well as strains of *P. acidilactici* where this operon is inactivated (since genetic tools have been developed for this bacterium).

While we agree that additional data often strengthens manuscripts, in this case, we felt it was unnecessary since we end up not using the biosynthetic peptides in favor of the chemically synthesized ones.

- In Figure 1C and 1E, please change the yellow color for *B. subtilis* since it is barely visible.

We chose this color palette since it was recommended as color-blind-safe used is from Nature Chem Biol. However, we agree that it can be hard to read – so we have changed the color palette – as shown below. We have also re-colored Figures 2, 3, and 4 to ensure consistency in color for species across all figures.

- For the conjugation of Pediocin, one would appreciate to have information on the efficacy and percentages of conjugation since the presence of non-conjugated pediocin in the used solution could strongly impact on the killing efficacy and binding measurements.

We thank the reviewer for this important point. We agree that the proportion of conjugated versus unconjugated pediocin could influence efficacy and binding outcomes. Based on Figure S7, we can conclude that very little of unconjugated peptide remains since the EC₅₀ increases >100-fold indicating that almost no unconjugated peptide remains.

- For figure S7 it is said that « However, the EC₅₀ was manifold lower relative to the unmodified peptide ». Please provide real numbers characterizing the difference of activity.

The text is modified as shown below. We would also like to acknowledge that there was mistake in the text since a lower EC₅₀ would indicate higher potency. Since the potency of the peptides decreased, we have now stated that the EC₅₀ has increased.

However, the EC₅₀ was **increased by >100-fold** relative to the unmodified peptide (**Figure S7**).

- Could the authors clarified in the text that they called the purified version of Pediocin from *L. plantarum* a Biosynthetic version of it ?

We have edited the manuscript to clarify this point. In the Results section “Purification of pediocin PA-1 yields partially purified peptide with significant impurities.”

Based on the degree of bioactivity, we concluded that the level of recombinant biosynthetic pediocin PA-1 accumulated in *L. plantarum* culture was similar to that observed for *P. acidilactici* UL5 and was therefore not pursued further

- For figure 2C, it is said : « and then quantified its EC50 of synthetic pediocin PA-1 against *L. seeligeri* (2.5 nM) and *L. coryniformis* (4.7 nM) ». I think this is wrong. The EC50s are 11 nM +/- 2.5 and 5 nM.

Apologies for the error. It is now corrected as:

We then confirmed its antimicrobial activity against using agar-well diffusion assay (**Figure 2B**) and then quantified its EC₅₀ of synthetic pediocin PA-1 against *L. seeligeri* (11 ± 2.5 nM) and *L. coryniformis* (11 ± 5 nM) (**Figure 2C**).

- Can the authors comment on the fact that the EC50 of the synthetic Ped is 10 fold higher than the purified one for *L. coryniformis* but the same for *L. seeligeri*.

Unfortunately, we do not have a good explanation for several observations relating to the behavior of the biosynthetic pediocin PA-1 peptide, which is why we stopped using it.

- In Figure S8 it could be helpful to indicate the localisation of the N-ter and C-ter part of the peptides, as well as, the parts described as responsible for the toxic vs binding activities.

We have updated the figure to include both details. See below:

- The K_d measurement of the Pediocin-FITC has been done in specific pH conditions. How, for example, the K_d of 82 nM to *L. seeligeri* compares to the one of the non conjugated Pediocin performed in the exact same conditions ?

Both experiments were performed under the same conditions, including pH.

- In relation with the experiments on synthetic communities, one can ask what would be happen if in the mix other *Listeria* species would be included. One of the main limitation of the use of bacteriocins is the fact that they can sometime target close species (which, by the way, question on their use in very complex communities where numerous species from the same genera are cohabiting).

This is a valid point – but we never claim species-level resolution. We describe the ability to target a specific subset of species in a community. In our work, pediocin PA-1 not only bind to *Listeria* sp. but also *L. coryniformis*.

- Also could the authors comment on their choice of using a 5 species community when testing killing activity while using a 6 species community (with the addition of *L. coryniformis*) when testing binding activity.

The difference between the two experiments is because conjugation alters the targeting specificity of pediocin. Specifically, unmodified pediocin PA-1 can target and kill both *L. seeligeri* and *L. coryniformis* (as highlighted in Figure 2C). For the killing study, we wanted to specifically target only 1 species in a co-culture, so we included only *L. seeligeri*. The FITC-conjugated peptides have a higher preference for *L. seeligeri* than *L. coryniformis* (indicated in Figure 3A, C), so we could safely include both species, demonstrating preferred enrichment of the latter.

- Why *L. coryniformis* is not depleted (but instead is enhanced) is the fluorescent-negative population since it is a target of Pediocin (Figure 4B) ?

This is indeed an interesting observation. This is because conjugation alters the targeting specificity of pediocin. Specifically, unmodified pediocin PA-1 can target and kill both *L. seeligeri* and *L. coryniformis* (as highlighted in Figure 2C). On the other hand, the FITC-conjugated peptides have a significantly higher preference for *L. seeligeri* than *L. coryniformis* (indicated in Figure 3A, C). Therefore, as expected, only *L. seeligeri* enriches in Figure 4B since we used conjugated peptides.

- Since there is an important debate on the use of probiotics versus their active compounds to control certain pathogenic bacteria, it would be very interesting to see how does *P. acidilactici* (and ideally its pediocin isogenic mutant) perform in killing experiments with complex communities compared to the purified or synthetic version of Pediocin.

This is an interesting point and well-worth studying. However, that is not within the scope of our work, which is focused on developing chemical biology tools to manipulate microbial communities through lethal and non-lethal binding.

- Could Table S1 provide the binding of pediocin to *L. seeligeri* ?

Unfortunately, we did not collect that data for the biosynthetic peptide in Figure 1C and we currently do not have the ability to evaluate it. However, given that the biosynthetic peptide did not behave as expected, the omission of this data does not change any of the conclusions drawn from this work.

- For the killing assay in the mixed community, how accurate and quantitative enough is the use of 16S DNA PCR and sequencing ? What about the presence of extracellular DNA coming from dead cells? Did the authors considered ddPCR and the use of for exemple PMA technology to ensure the absence of amplification from dead cells?

This is an interesting point – but we do not expect significant contamination from extracellular DNA. Since we used pelleted and thoroughly washed cells prior to 16S sequencing, we expect that DNA from lysed cells in the culture media would largely be washed away.

Reviewer #2 (Comments for the Author):

This is an exciting manuscript that presents a valuable tool for the microbiome research community. If anything, the work may be underselling the broader potential of this approach. I look forward to implementing this technique in my own research and expect others will be equally enthusiastic. The authors may wish to further emphasize the proof-of-concept nature of this work and its potential for diverse applications.

Thank you for the kind words and appreciation for this work.

Overview

This manuscript presents a clever approach for targeted microbiome manipulation using narrow-spectrum class II peptide bacteriocins. The authors demonstrate this technique using pediocin PA-1 as a proof-of-concept, and the methodology could be readily adapted to other natural bacteriocins for various applications.

Thank you for the kind words.

Major Comments

1. The Discussion section would benefit from addressing how this bacteriocin-based approach compared to other methods for specific microbiome labeling or manipulation. Consider discussing the relative advantages, limitations, and potential use cases of this technique compared to alternative approaches such as CRISPR-based methods.
2. The Discussion should more strongly emphasize the significance of this technique's dual capability: (1) specifically labeling pediocin-sensitive species without killing them, and (2) selectively eliminating specific community members. Both applications have compelling use cases!
3. Please address potential limitations when adapting this technique to other systems. Specifically, successful implementation in an undefined community requires: (1) identification of an appropriate bacteriocin that targets the organism of interest, and (2) comprehensive knowledge of community composition and bacteriocin-specificity to predict off-target effects. While this doesn't diminish the utility of this approach, it does mean that the technique would require quite a bit of consideration to transfer to new systems.

Thank you for these suggestions. We have completely re-written our discussion section to address the issues raised above. Please see below:

In **this** study we evaluated the antimicrobial spectrum, potency, and potential applications of pediocin PA-1 **for manipulating microbial communities**. Consistent with prior reports, the bioactivity of the crude supernatant of *P. acidilactici* UL5 was limited to *Listeria* species and *Lactobacillus coryniformis*, with no detectable inhibition of other lactic acid bacteria or Gram-negative *E. coli* [42]. **While we had some success with using the purified, biosynthetic pediocin**

PA-1, chemical synthesis provided an alternate route to obtaining a product with expected properties – correct molecular weight, narrow-spectrum activity, and EC₅₀ values in the low nanomolar range against *L. seeligeri* and *L. coryniformis*.

Prior to this work, bactericidal activities of pediocin PA-1 and other narrow-spectrum bacteriocins have been well-explored to control contamination in food [24–28], kill pathogens [29–34], and in studying the role of probiotics and their derivatives on microbial communities [29, 35–41]. But they have not been well-explored as tools to manipulate mixed microbial systems. Our results corroborate the established narrow-specificity of pediocin PA-1 and demonstrated targeted killing in a mixed microbial system, highlighting their use for targeted depletion of a subset of species within communities. While other approaches have used phages [53–55], lytic enzymes [21, 56], or intracellular toxins (including CRISPR/Cas) expression [22, 23, 57, 58] to deplete small numbers of species, they have limitations that our approach may overcome. For example, phages are not well-known for many species and identifying new phages is not trivial. Lytic enzymes are highly labile and readily inactivated or degraded. Intracellular delivery of toxic payloads poses significant hurdles in ensuring high-efficiency and targeted delivery and may require extensive engineering of conjugative, phage, or other gene-transfer systems. Further, addition of engineered species with payloads could drastically alter community dynamics, making them more suitable for applied and/or translational studies rather than those on basics of microbial ecology. Comparatively, our peptide-based approach is simpler since bacteriocins against a wide-variety of species have already been identified and can be mined from metagenomic sequences [59–63]. Additionally, the ability to chemically synthesize peptide bacteriocins can make testing and validation quicker. Finally, peptide bacteriocins are generally less prone to denaturation and since

they kill cells extracellularly without transcription/translation steps, they can alter community composition more rapidly.

Beyond killing, we also demonstrated that modified peptides derived from pediocin PA-1 (whether conjugated to functional handles or truncated) maintain their target specificity while losing their bactericidal activity. Such reagents open the opportunity to manipulate mixed microbial systems from the top-down in a non-lethal way to enable fundamental studies with extant microbial communities. This includes, but is not limited to, elucidating roles of specific species in establishment, evolution, and resilience of ecologies. Current approaches suffer from similar limitations as those described for killing – low specificity, slow response times, and complexity [19, 22, 64, 65].

Our approach is not without limitations, however. We found that the peptides show pH-dependent activity, which may affect their activity under uncontrolled conditions (e.g., in situ). Our method development was supported by extensive published literature on the spectrum, potency, activity, and structure-function relationships of pediocin PA-1, which may not be readily available for other peptides. Thus, as with any method involving targeted manipulation of specific species, initial characterization will be needed prior to deployment.

Nonetheless, the potential utility of our approach is amplified by its overall simplicity. Since we used a naturally-occurring bacteriocins and made small modifications, the approach could readily be adapted to other bacteriocins to manipulate other microbes and communities. With potential for multiplexing, this approach may provide a powerful path forward to advancing basic and applied

science and engineering microbiota for biomedical, veterinary, agricultural, biomanufacturing, and environmental applications.

Minor Comments

Abstract

- Two consecutive sentences begin with “Finally”. Please rephrase one to improve readability.

We have edited as:

Next, we conjugate chemical handles on the bacteriocin and show that the binding spectrum is largely unchanged. **Then**, using truncated variants, also conjugated to chemical handles, we show functional non-bactericidal binders that largely maintain their specificity. Finally, with the unmodified and modified bacteriocins, we show that specific bacteria can be depleted through killing or cell sorting within a mixture of highly similar bacteria.

Introduction, Paragraph 2

- Is the paragraph intended to begin with “But”?
- Consider revising “sequencing-based studies” to “sequencing-based approaches.”

“But” appears twice so we have removed one instance. We have also replaced studies with approaches.

Widely used sequencing-based **approaches** can correlate presence of species with community phenotypes – but they are largely hypothesis generating.

- The rationale for needing additional microbiome manipulation tools is valid, but the connection could be clarified. Please explicitly explain how tools enabling targeted community alteration support the testing of hypotheses generated by traditional/sequence-based approaches.

Edited.

To test these hypotheses and to glean a deeper understanding of microbial communities, engineering tools and approaches are needed to **non-destructively** alter community composition.

- Final sentence: Consider changing “prevents targeted manipulation” to “limits targeted manipulation.”

Fixed.

However, there is a technological gap that **limits** targeted manipulation of subpopulations within communities.

Results, Paragraph 2

- The authors state they cloned the “pediocin operon” into pSIP-P27 but do not define how this operon was delineated. Has this operon been functionally characterized previously? If so, please provide a reference. If not, please describe how the cloned region was determined.

We have edited the text:

We then proceeded to clone the pediocin operon (*pedABCD* [43]) from *P. acidilactici* UL5 into pSIP-P27 vector for heterologous expression in other lactic acid bacteria viz. *Lactiplantibacillus plantarum* WCFS1, *Lactococcus cremoris* MG1363 and the well-established expression host, *E. coli* MG1655.

Methods

- The description of agar overlay assays is missing from the Methods section. Please include this protocol.

We apologize for the omission. We have added it to the methods sections.

Agar overlay assay.

The antimicrobial activity of pediocin was assessed using an SDS–PAGE gel-overlay assay [68]. Protein samples were resolved on a 4–20% acrylamide/bis-acrylamide SDS–PAGE gel, after which the gel was thoroughly washed with sterile water to remove residual SDS. The gel was then placed onto BHI (brain heart infusion) agar plates and overlaid with 0.8 % BHI soft agar inoculated with *L. seeligeri*. Following incubation at 37 °C overnight, plates were examined for zones of growth inhibition indicative of pediocin activity. The bactericidal activity of chimeric proteins was analyzed by polyacrylamide gel electrophoresis and gel overlay screening. Cell pellet and culture

supernatant samples first were run on 16 % acrylamide-bis-acrylamide–SDS gels. Subsequently, the gels were washed in sterile water for 3 h to remove SDS, placed on pre-poured TGE (Tryptone Glucose Extract) agar plates, and covered with a 20 mL TGE soft-agar overlay containing *L. seeligeri* cells. The plates were incubated at 37 °C overnight and examined for zones of growth inhibition associated with proteins in the samples.

Re: Spectrum03215-25R1 (**Engineered antimicrobial-derived peptides to manipulate mixed microbial systems**)

Dear Prof. Nair:

Thank you for the privilege of reviewing the revised version of your manuscript. Below you will find the reviewer's suggestions, which should be easy to address.

Revision Guidelines

Sincerely,
Sergey Zotchev
Editor
Microbiology Spectrum

Reviewer #2 (Comments for the Author):

The authors have made significant changes in the manuscript that strengthen this work quite nicely. In particular, the discussion is much stronger and more appropriately references the existing literature. With the exception of a few minor points described below, the authors have addressed all previous reviewers' comments.

1. Figure 1: Some of the figure references are mixed up in the main text. Line 132 does not seem to refer to Fig 1B; Line 135 "1B" should be replaced with "1C".
2. Line 213: Can the authors show a range of tested pHs here? "...binding to *L. seeligeri* at all the tested pHs [range] but found..."
3. Line 236: Clarify either the text or the figure here. Text states 4h, indicating 4h pediocin treatment. Figure label states 6h, indicating 6h culture growth (including the 2h growth period before pediocin was added).
4. Regarding this point from Reviewer 1, below. This caveat could be more explicitly stated in the discussion. This is indeed an inherent limitation of bacteriocins, and the authors could address this by stating that future work is necessary to understand the species and strain-level specificity that constrain target range.

Reviewer 1 Comment: In relation with the experiments on synthetic communities, one can ask what would be happen if in the mix other *Listeria* species would be included. One of the main limitation of the use of bacteriocins is the fact that they can sometime target close species (which, by the way, question on their use in very complex communities where numerous species from the same genera are cohabiting).

Responses to reviewers are in blue. Updated manuscript text is in red.

Reviewer #2 (Comments for the Author):

The authors have made significant changes in the manuscript that strengthen this work quite nicely. In particular, the discussion is much stronger and more appropriately references the existing literature. With the exception of a few minor points described below, the authors have addressed all previous reviewers' comments.

1. Figure 1: Some of the figure references are mixed up in the main text. Line 132 does not seem to refer to Fig 1B; Line 135 "1B" should be replaced with "1C".

Apologies, we have corrected this now.

2. Line 213: Can the authors show a range of tested pHs here? "...binding to *L. seeligeri* at all the tested pHs [range] but found..."

We have added this detail to the text now:

We observed that pediocin-FITC showed strong binding to *L. seeligeri* at all the tested pHs (4.0 – 7.4) but found surprisingly poor binding to *L. coryniformis* under acidic conditions (Figure S10).

3. Line 236: Clarify either the text or the figure here. Text states 4h, indicating 4h pediocin treatment. Figure label states 6h, indicating 6h culture growth (including the 2h growth period before pediocin was added).

We apologize for the confusion. The 4 h refers to time elapsed since peptide addition, which is at 6 h (= 2 + 4). We can see how this can be confusing. We have edited the text for clarity:

We observed that after 4 h (i.e., 6 h from inoculum), the proportion of *L. seeligeri* in the control (untreated) increased from 18 % to 60 %.

4. Regarding this point from Reviewer 1, below. This caveat could be more explicitly stated in the discussion. This is indeed an inherent limitation of bacteriocins, and the authors could address this by stating that future work is necessary to understand the species and strain-level specificity that constrain target range.

Please see our response to the comment below.

Reviewer 1 Comment:

In relation with the experiments on synthetic communities, one can ask what would be happen if in the mix other *Listeria* species would be included. One of the main limitation of the use of bacteriocins is the fact that they can sometime target close species (which, by the way, question on their use in very complex communities where numerous species from the same genera are cohabiting).

We agree with the reviewer that the bacteriocin-based approach described here may not provide species-level specificity. Indeed, this is apparent from our results that show that pediocin PA-1 targets both *L. seeligeri* and *L. coryniformis*. In addition, we acknowledge this in various sections of the manuscript and refrain from stating that our approach provides species-level specificity.

In the Abstract, we state, “In this work, we develop molecular probes to manipulate **specific subpopulations** within multispecies microbial populations...”

In the Introduction we state, “we leverage natural and engineered bacterially-derived antimicrobial peptides (AMPs) called bacteriocins as means to manipulate **specific members** within a mixed microbial system”, “We show that pediocin PA-1 can be used to specifically deplete **few members** within a mixed community of highly similar bacterial species”, “We also demonstrate that pediocin PA-1 can be engineered to serve as a non-bactericidal binding reagent that specifically tags and manipulates **a subset of species** in a culture of phylogenetically similar species.”

In the Discussion, we use the terminology, “Our results corroborate the established narrow-specificity of pediocin PA-1 and demonstrated targeted killing in a mixed microbial system, highlighting their use for targeted depletion of a **subset of species** within communities.”.

Nonetheless, we have added a statement to further emphasize this point in the manuscript in the limitations paragraph of the Discussion:

Our approach is not without limitations, however. We found that the peptides show pH-dependent activity, which may affect their activity under uncontrolled conditions (e.g., in situ). **Additionally, the peptides do not have absolute specificity, resulting in potential targeting of multiple species – such as various *Listeria sp.* and *L. coryniformis* by pediocin PA1.** Our method development was supported by extensive published literature on the spectrum, potency, activity, and structure-function relationships of pediocin PA-1, which may not be readily available for other peptides. Thus, as with any method involving targeted manipulation of specific species, initial characterization will be needed prior to deployment.

Re: Spectrum03215-25R2 (**Engineered antimicrobial-derived peptides to manipulate mixed microbial systems**)

Dear Prof. Nair:

I am pleased to inform you that your manuscript has been accepted, and it is being forwarded to the ASM production staff for publication. Your paper will first be checked to make sure all elements meet the technical requirements. ASM staff will contact you if anything needs to be revised before copyediting and production can begin. Otherwise, you will be notified when your proofs are ready to be viewed.

Thank you for submitting your manuscript to Spectrum.

Best regards,
Sergey Zotchev
Editor
Microbiology Spectrum